# CHD4 slides nucleosomes by decoupling entry- and exit-side DNA translocation

Yichen Zhong[1,6], Bishnu P. Paudel [2,3,6], Daniel P. Ryan[4], Jason K.K. Low[1], Charlotte Franck[1,5], Karishma Patel [1], Max J. Bedward[1], Mario Torrado[1], Richard J. Payne[5], Antoine M. van Oijen [2,3✉] & Joel P. Mackay [1✉]

Chromatin remodellers hydrolyse ATP to move nucleosomal DNA against histone octamers. The mechanism, however, is only partially resolved, and it is unclear if it is conserved among the four remodeller families. Here we use single-molecule assays to examine the mechanism of action of CHD4, which is part of the least well understood family. We demonstrate that the binding energy for CHD4-nucleosome complex formation—even in the absence of nucleotide—triggers significant conformational changes in DNA at the entry side, effectively priming the system for remodelling. During remodelling, flanking DNA enters the nucleosome in a continuous, gradual manner but exits in concerted 4–6 base-pair steps. This decoupling of entry- and exit-side translocation suggests that ATP-driven movement of entry-side DNA builds up strain inside the nucleosome that is subsequently released at the exit side by DNA expulsion. Based on our work and previous studies, we propose a mechanism for nucleosome sliding.

[1] School of Life and Environmental Sciences, University of Sydney, Sydney, NSW 2006, Australia. [2] Molecular Horizons, School of Chemistry and Molecular Bioscience, University of Wollongong, Wollongong, NSW 2522, Australia. [3] Illawarra Health and Medical Research Institute, Wollongong, NSW 2522, Australia. [4] Department of Genome Sciences, The John Curtin School of Medical Research, The Australian National University, Canberra, ACT 2601, Australia. [5] School of Chemistry, The University of Sydney, Sydney, NSW 2006, Australia. [6]These authors contributed equally: Yichen Zhong, Bishnu P. Paudel. ✉email: vanoijen@uow.edu.au; joel.mackay@sydney.edu.au

Nucleosomes are made up of a histone octamer containing two copies of each histone (H2A, H2B, H3 and H4) wrapped by 147 base pairs (bp) of DNA[1] (Fig. 1a). Through the activity of chromatin modifying enzymes, nucleosomes regulate DNA accessibility and therefore gene expression.

Chromatin remodellers are specialized ATP-driven translocases that reposition, eject, and replace histones within the nucleosome. These events happen in response to stimuli such as epigenetic modifications and cell cycle signals[2], thus exposing the DNA to (or protecting it from) other DNA-binding or transcription regulatory factors[3]. Mutation or dysfunction of chromatin remodellers often leads to severe consequences, including disrupted cell cycle, tumorigenesis and early embryonic lethality[4].

All known chromatin remodellers are superfamily 2 (SF2) ATPase motors, which consist of two lobes that form an active-site cleft[5,6]. However, the domains flanking the ATPase vary and are used to classify chromatin remodellers into four families: SWI/SNF (mating type switching/sucrose non-fermenting), ISWI (imitation switch), INO80 (inositol), and CHD (chromodomain

**Fig. 1 CHD1/CHD4 domain topology and nucleosome repositioning assays. a** Schematic showing the 15 superhelical location (SHL) sites, the most outward-facing positions of the minor groove, in the 601 nucleosome positioning sequence (labelled +7 to −7). The histone–DNA interface consists mainly of inward-facing sections of the DNA minor groove, which are defined as superhelical locations (SHL) ± 0.5–6.5[31]. Orange bands indicate four phased TpA dinucleotides that are spaced 10 bp apart. **b** Domain architectures of yeast CHD1 and human CHD4 with residues at domain boundaries indicated (NTD: N-terminal domain, PHD: plant homeodomain, CHCT: CHD1 C-terminal domain). **c** Gel-based nucleosome repositioning assays carried out with the indicated nucleosomes and remodellers. Fluorescently labelled nucleosomes were treated with the indicated remodeller for 60 min, the reaction was stopped by adding dsDNA (33 µg/mL) and then the samples were run on 5% native polyacrylamide gels. Symmetrically positioned nucleosomes are retarded relative to asymmetrically positioned species, as indicated. **d** Gel-based nucleosome repositioning assays carried out as described in (**c**), except that an incubation time-course was carried out at a single CHD4 concentration (5 nM). Assays using 0w60 (upper panel) or 30w30 (lower panel) substrates are shown; a 0w60 control lane was included in the lower panel. Source data are provided as a Source Data file. **e, f** Nucleosome sliding assays carried out as described in (**c**), using the indicated nucleosome substrates and 5 nM CHD4. Remodelled products (0w30 and 30w0) are indicated by red arrows, and the possible hexosome band is indicated by black arrows. Source data are provided as a Source Data file.

helicase DNA-binding). Uniquely, CHD remodellers possess two tandem chromodomains adjacent to the ATPase. Some CHD remodellers can function as a monomer, such as yeast CHD1[7], but metazoan CHDs generally exist in multi-protein complexes. The CHD family member CHD4 is a subunit of the nucleosome remodelling and deacetylase (NuRD) complex[8], a complex that also contains the histone deacetylases HDAC1 and -2.

The mechanism by which these enzymes remodel nucleosomes remains ambiguous. Previous studies show that ISWI, SWI/SNF and CHD1 remodellers bind to nucleosomal DNA at a position two helical turns from the dyad, and remain at this position during the remodelling process[9–12]. Single-molecule Förster resonance energy transfer (smFRET) based measurements using different remodellers have suggested that DNA exits the nucleosome in short 1–7 bp segments[13–15]. The behaviour of DNA at the entry side is less well understood; bidirectional or reversible remodelling of CHD1 and ISWI has even been proposed, based on observations of fast direction-switching of DNA movement[14,16]. Overall, it has been hard to correlate entry- and exit-side behaviour and multiple models have been proposed for the remodelling mechanism. Of these, two have become most prominent[17,18]. The "DNA loop/wave propagation" model states that the DNA entering the nucleosome from the entry side forms a ~10-bp bulge/loop on the octamer surface that quickly propagates around the histone core and is subsequently released from the exit site. The "twist diffusion" model proposes that the remodeller changes the structure of the DNA helix, causing a twist defect, and hydrolysis of ATP results in directional transfer of a few base pairs to the adjacent DNA segment[19,20]. Both models hypothesise that the entry site movement happens prior to DNA exiting the nucleosome, a concept that is supported by a recent three-fluorophore smFRET analysis of CHD1 and ISWI activity[21].

In this study, we investigate the mechanism by which CHD4 remodels mononucleosomes. We combine gel-based assays and smFRET methods to monitor the nucleosome sliding activity of CHD4 in real time. The data show that during remodelling, DNA enters the nucleosome continuously but exits in bursts of 4–6-bp, demonstrating that the entry and exit processes are partially decoupled. We also show that CHD4 binding in the absence of nucleotide induces substantial dynamics at the DNA entry side, suggesting that the binding energy of CHD4 alone makes a significant contribution to remodelling. Overall, our results reveal mechanistic aspects of the process by which CHD4 re-organizes nucleosomes in the genome.

## Results

### CHD4 remodels both symmetric and asymmetric nucleosomes.
We first assessed the nucleosome sliding activity of human CHD4 using a gel-based remodelling assay. We expressed and purified human histones H2A, H2B, H3 and H4 and reconstituted nucleosomes using the 601 Widom positioning DNA sequence[22] to which a 60-bp extension was added at the +7 end (Fig. 1a). We refer to this construct as 0w60. Following incubation with CHD4 and ATP, the products were run on a native gel. New bands, which have been previously shown to be more symmetrically arranged nucleosome[23,24], were observed (Fig. 1c, left panel). Although the intensity of the upper bands increased at higher CHD4 concentrations and with longer incubation times (Fig. 1d, top panel), we still observed ~50% starting material. Similar results were observed at 25 °C (Supplementary Fig. 1a). We repeated the remodelling assay using the same amount of native NuRD complex, which contains additional HDAC, MTA, RBBP, MBD and GATAD2 subunits. An identical pattern of shifted bands was observed (Fig. 1c, middle panel), indicating that

the additional subunits do not significantly alter the product distribution, in contrast to the observation made previously for the CHRAC complex[25].

To investigate the effect of substrate conformation, and to determine if CHD4 could reversibly remodel its products, we also assessed the activity of CHD4 towards a nucleosome bearing 30-bp extensions on both sides (30w30). In this case, ~80% of the starting material was converted to less symmetric products (Fig. 1c, right panel, and Fig. 1d, lower panel), indicating that CHD4 can remodel mononucleosomes with either symmetric or asymmetric extensions, but displays a substrate preference and greater remodelling efficiency towards the former.

The 601 Widom sequence used for our nucleosomes is non-palindromic. The minus gyre (Fig. 1a) contains four phased TpA dinucleotides at inward-facing minor groove positions, providing extra DNA-nucleosome stability[26,27]. Therefore, we reconstituted a 60w0 construct so that the histone octamer would be remodelled towards the minus end of the 601 sequence. Like its 0w60 counterpart, 60w0 was partially but not fully remodelled. With the extension now located on the 'minus' side, closest in sequence to the TA-rich region, the final set of products displayed a similar but not identical distribution (Fig. 1e). The bands with intermediate migration probably represent partially remodelled nucleosomes. We also observed similar changes in remodelling assays using nucleosomes containing the MMTV nucA positioning sequence (Supplementary Fig. 1b), consistent with the CHD1 remodelling results seen for the MMTV nucA nucleosome[28]. These data suggest that CHD4 has some ability to discriminate between substrates with different DNA sequences.

Next, we tested the effect of the extra-nucleosomal DNA length on CHD4 remodelling activity. Remodellers such as INO80 and ISWI can sense the length of extra-nucleosomal DNA and prefer substrates with longer flanking DNA[9,29]. CHD4 was able to remodel nucleosomes with a 30-bp extension (0w30 and 30w0) (Fig. 1f), although the extent of remodelling was significantly less than for 0w60. A lower-migrating band also appeared after incubation and, based on published data[30], is likely to be a hexasome bearing a single H2A-H2B dimer. We are unsure whether the hexasome is a by-product or rather represents a novel activity of CHD4. A 30w30 substrate showed significantly more remodelling and less hexasome formation (Fig. 1e), suggesting that total DNA length might impact on remodelling behaviour and nucleosome stability during remodelling.

### Nucleosome binding by CHD4 is independent of flanking DNA.
Because our data suggested that a symmetric nucleosome with longer flanking DNA is a more favourable substrate for CHD4, we sought to test whether this selectivity arises from DNA-binding preferences of the protein. We therefore bound FLAG-CHD4 to anti-FLAG beads and incubated with 0w0, 0w60, 30w30 or 60w60 nucleosomes. Surprisingly, CHD4 pulled down all constructs with no clear preference for the flanking DNA lengths and/or substrate symmetry (Fig. 2a), suggesting the differences in remodelling activity are independent of substrate binding selectivity.

Electrophoretic mobility shift assays (EMSAs) were then used to quantify DNA-binding affinity. Figure 2b shows that 0w0, 0w30 and 0w60 all bind CHD4 with comparable affinity with dissociation constants of ~40–80 nM. The fact that even the nucleosome with no flanking DNA (0w0) displayed a similar affinity to the other constructs suggests that a significant number of the contacts made by CHD4 are to the nucleosome core particle rather than to flanking DNA. Furthermore, binding of a second CHD4 molecule can be observed at high CHD4 concentration, indicating that two CHD4 binding sites exist and

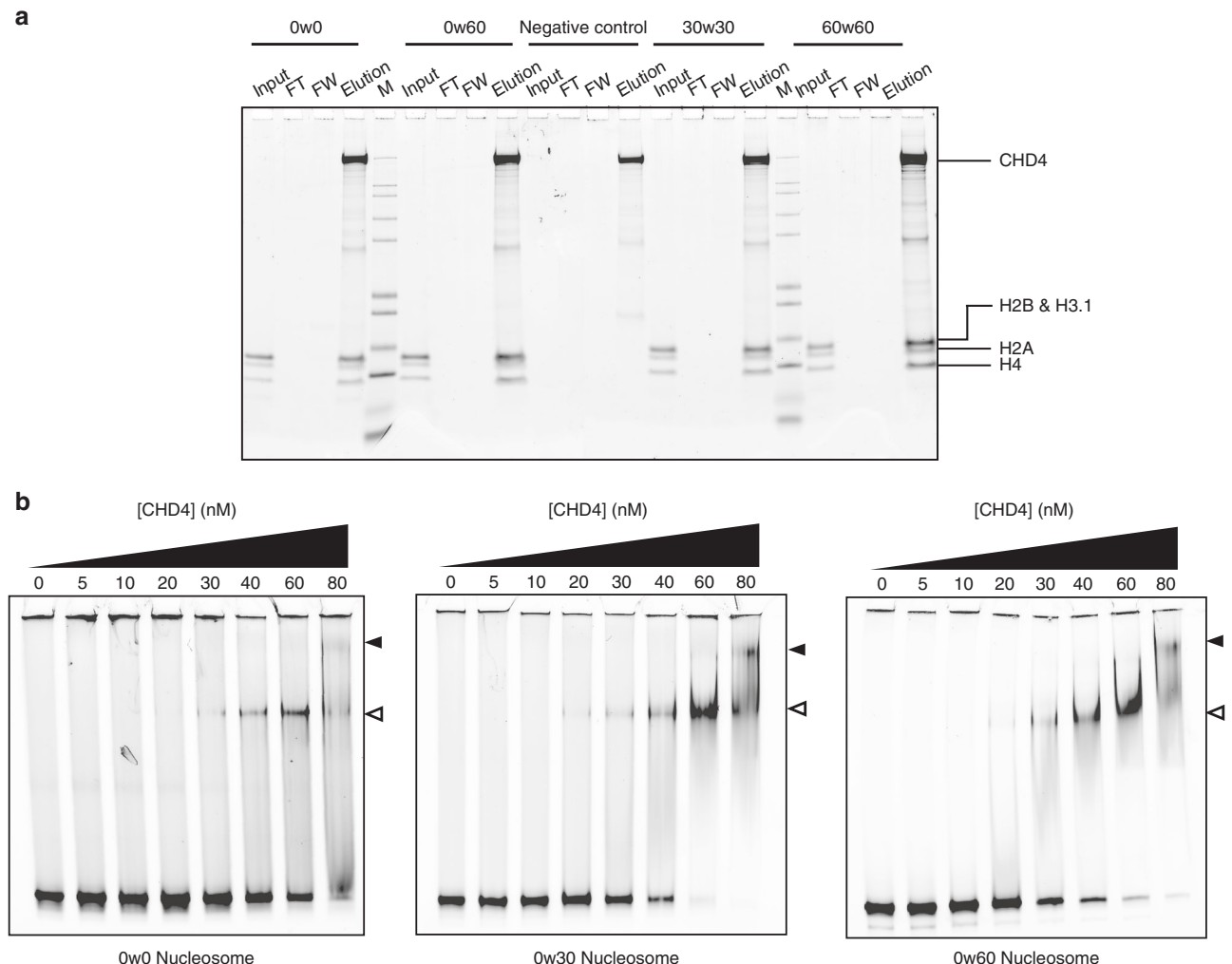

**Fig. 2 The affinity of CHD4 for DNA is not strongly dependent on the presence of flanking extra-nucleosomal DNA. a** Pulldowns in which FLAG-CHD4 is immobilized on FLAG-Sepharose beads and incubated with the indicated nucleosomes, followed by elution with FLAG peptide. Nucleosome input, unbound nucleosome/flow through (FT), final wash (FW) and elutions of each nucleosome construct were analysed by SDS-PAGE, along with a size indicator, Mark12 Standard (M). The negative control contains no nucleosome in the input. **b** EMSAs showing the binding of CHD4 to the indicated nucleosomes. Nucleosome concentration was 90 nM (0w0) or 60 nM (0w30 and 0w60). Open and filled triangles indicate complexes that probably contain one or two CHD4 molecules, respectively.

that these two sites must lie sufficiently far from the dyad axis of the nucleosome that they can be co-occupied. The pattern of shifts also indicates that the two CHD4 binding events are not cooperative.

**CHD4 ejects DNA from the nucleosome in multi-base-pair steps.** To further understand the mechanism of CHD4-driven nucleosome sliding, we used smFRET to monitor nucleosome conformational changes. The ability to follow a single nucleosome through time allows the observation of transient remodelling intermediates and thus provides access to mechanistic details of the sliding process. We assembled nucleosomes using DNA tagged with a 5′ donor dye (AlexaFluor555, AF555) at one end and with 5′ biotin at the other (Fig. 3a). We used the well-studied T120C mutant of H2A in these experiments, which was coupled to the acceptor dye AlexaFluor647 (AF647) via the cysteine. We optimised the labelling reaction to be incomplete (50% of H2A labelled) so that most nucleosomes contained only one copy of labelled H2A that is either proximal or distal to the AF555 DNA tag. We named these constructs $n^{AF555}w60^{Bio}$, where $n$ indicates

the number of base pairs between the AF555 and the 601 sequence. Because our gel-based experiments suggested that CHD4 remodels 0w60 to a more symmetric conformation, we refer the end with the 60-bp flanking DNA as the entry site, and the other as the exit site. The nucleosome substrates were immobilized on a coverslip (Fig. 3a) and embedded in a microfluidic channel, and the coverslip was imaged using total internal reflection fluorescence (TIRF) microscopy. The AF555 and AF647 form a FRET pair, which reports on the apparent distance between the DNA and histone labels.

Imaging of immobilized $0^{AF555}w60^{Bio}$ nucleosomes yielded three populations with distinct FRET values (note that in the text that follows, 'FRET' refers to FRET efficiency), which correspond to nucleosomes bearing AF647-H2A at either the proximal or the distal site, relative to AF555, or both (Fig. 3b). Unless indicated, only the mid-FRET population that contains a single, proximal label was selected. To establish the behaviour of this system as a function of time, we first monitored the FRET of the immobilized $0^{AF555}w60^{Bio}$ nucleosomes alone. Figure 3c shows that no significant change in FRET is observed until one of the fluorophores undergoes photobleaching. Nucleosomes bearing

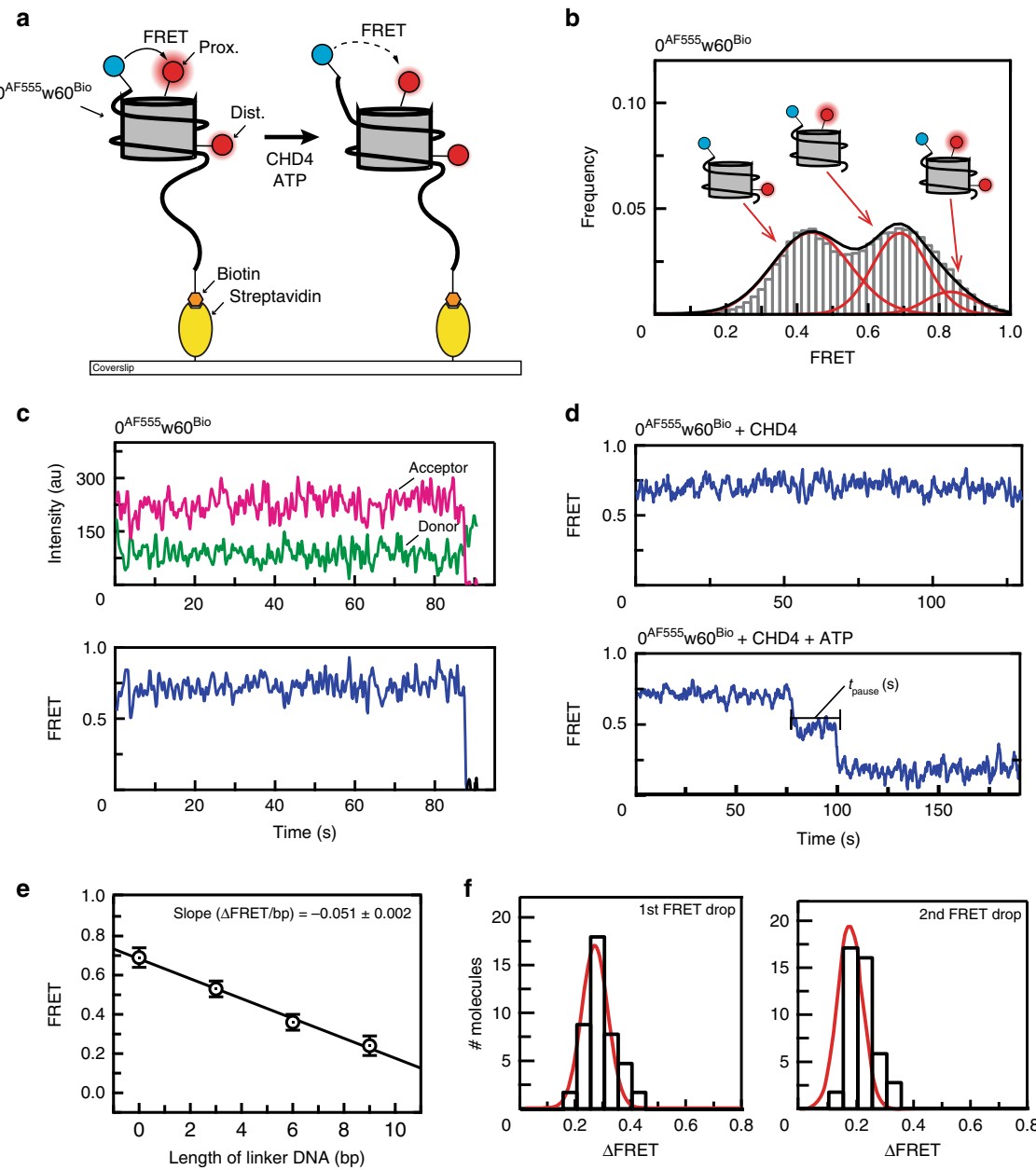

**Fig. 3 A single-molecule FRET (smFRET) assay for nucleosome sliding shows that CHD4 drives multi-base-pair movements of nucleosomal DNA at the exit side. a** Diagram depicting the setup for the smFRET assay and nucleosome substrate before and after remodelling by CHD4. A $0^{AF555}w60^{Bio}$ nucleosome containing either proximal or distal or both AF647-labelled H2A is assembled onto a PEGylated coverslip via biotin at the longer end of the flanking DNA (AF555 and AF647 are represented by blue and red circles, respectively). FRET is then monitored as a function of time under different conditions. **b** Pre-reaction distribution of nucleosomal FRET states for $0^{AF555}w60^{Bio}$ nucleosomes. Low, mid- and high-FRET states correspond to particles containing H2A in the distal, proximal or both positions (relative to the DNA-bound AF555), respectively. **c** FRET vs time trace of $0^{AF555}w60^{Bio}$ nucleosome bearing a proximal H2A label. Donor AF555 fluorescence (green), acceptor AF647 fluorescence (magenta) and FRET (blue) are shown. **d** FRET vs time traces of $0^{AF555}w60^{Bio}$ nucleosome bearing a proximal H2A label in the presence of 2 nM CHD4 (top) or both 2 nM CHD4 and 1 mM ATP (bottom). Two clear drops in FRET are observed in the latter. We define the pause time $t_{pause}$ as the duration between two FRET changes. **e** Calibration of FRET values for $n^{AF555}w60^{Bio}$ nucleosomes. The FRET of proximally labelled particles was measured as a function of the number of base pairs ($n$) added to linker DNA at the exit site and mid-FRET peak values for each construct were obtained by fitting to a Gaussian distribution. Plotting the change in mid-FRET value as a function of $n$ yielded a slope of $-0.051 \pm 0.002$. Error bars represent standard deviation of the fit from at least two independent measurements. **f** Distribution of the 1st and 2nd step sizes for 40 molecules undergoing remodelling in presence of 2 nM CHD4 and 1 mM ATP. The histograms are fitted to a Gaussian distribution.

both proximal and distal labels undergo two steps of photo-bleaching (Supplementary Fig. 1c). Similarly, introducing CHD4 alone has no effect (Fig. 3d), indicating CHD4 binding does not induce any significant conformational changes in the nucleosome that alter the distance between the two fluorophores.

Next, we treated $0^{AF555}w60^{Bio}$ nucleosomes with both CHD4 and ATP in a continuous flow manner and monitored FRET as a function of time. A stepwise reduction in FRET was seen, indicating that the two fluorophores had moved further apart (Fig. 3d and Supplementary Fig. 1d). The traces consistently

showed two sharp FRET drops, separated by a pause time ($t_{pause}$) of constant FRET. In some cases, transient excursions to lower FRET values for up to ~10 s were observed (Supplementary Fig. 1e), followed by a return to the pre-excursion value. A similar overall pattern of two drops in FRET was observed for particles containing a distally labelled H2A (Supplementary Fig. 1f).

To correlate the observed changes in FRET to translocation of a particular number of DNA base pairs, we reconstituted a series of $n^{AF555}w60^{Bio}$ nucleosomes with $n = 0, 3, 6$ or $9$ and constructed a calibration curve (Fig. 3e), similar to previous studies[16]. These data indicated that a reduction in FRET of 0.051 ± 0.002 units corresponds to a 1-bp change in AF555 position relative to AF647. The two drops in Fig. 3d therefore correspond to 6- and 4-bp translocations (for the first and second steps, respectively) of the DNA end away from the core particle. These step sizes are consistent among independent remodelling events (Fig. 3f). The reason for the smaller second step size is currently unclear; it is possible that the state of the nucleosome following one round of remodelling is different to the starting state. Notably, both FRET changes remain as a single step at lower ATP concentrations, rather than breaking into smaller substeps (Supplementary Fig. 1g). As the DNA helix is around 10 bp per turn, these step sizes suggested that roughly a half-turn of DNA moves out of the nucleosome at each step and that, after two steps, a full turn of DNA will be translocated.

**The pause time $t_{pause}$ is ATP concentration dependent.** To probe the mechanism of remodelling, we next assessed the dependence of the pause time between the two remodelling events ($t_{pause}$, Fig. 3e) on reaction conditions. We first found that a 100-fold change in CHD4 concentration (from 200 pM to 20 nM) resulted in only a small increase in $t_{pause}$ (from 0.7 to 0.8 s, Fig. 4a), suggesting that the remodelling process does not rely on dissociation and rebinding of CHD4.

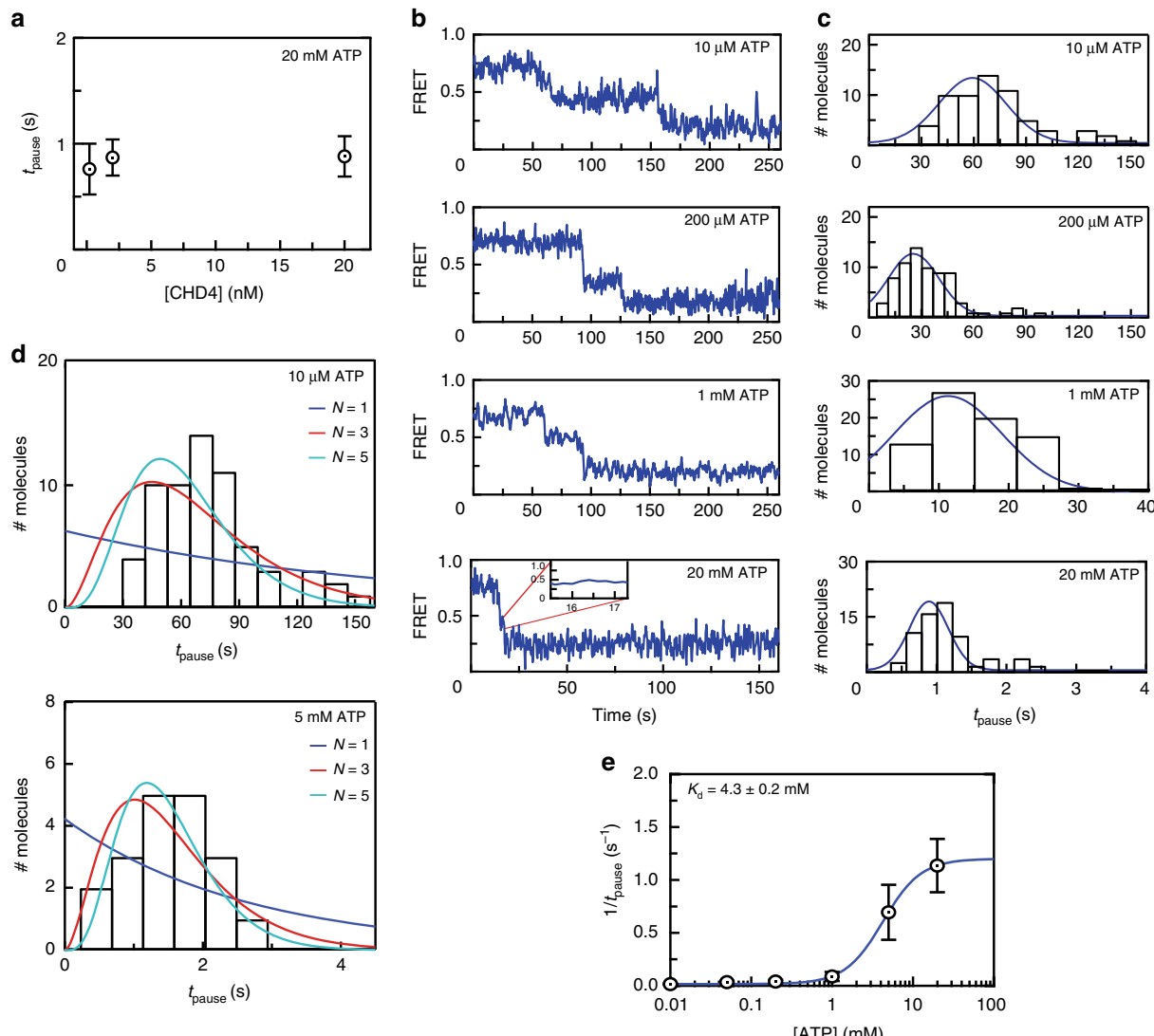

**Fig. 4 CHD4-mediated nucleosome sliding is processive and depends on the binding of multiple ATP molecules. a** Distribution of $t_{pause}$ times for remodelling of $0^{AF555}w60^{Bio}$ at the indicated CHD4 concentrations and 20 mM ATP. Error bars represent standard deviation (of 45, 74 and 56 independent single particles at 0.2, 2 and 20 nM CHD4, respectively). **b** Typical traces from smFRET assays carried out with 2 nM CHD4 and at the indicated ATP concentrations. **c** Histograms showing $t_{pause}$ distributions of 70–82 particles from the experiments shown in (**b**). The histograms are fitted to a Gaussian distribution. **d** Pause time histograms for experiments carried out at 10 µM ATP (top) or 5 mM ATP (bottom) are overlaid with gamma distributions depicting different numbers of fundamental reaction steps ($N = 1$–5). **e** A 1:1 binding isotherm fit of the mean $t_{pause}$ time as a function of ATP concentration, with data taken from the assays in (**c**). Error bars represent standard deviation.

In contrast, the reaction was strongly dependent on ATP concentration. Increasing ATP concentration from 10 µM to 20 mM reduced $t_{pause}$ by roughly 60-fold, to the point where the separation between two FRET drops was almost unobservable (Fig. 4b, c). Thus, under these conditions, ATP binding (and probably hydrolysis) is the rate-limiting step of the reaction. Importantly, if $t_{pause}$ comprises a single fundamental reaction step, then the distribution of $t_{pause}$ values should follow an exponential function. If the pause period instead comprises several sequential reactions, $t_{pause}$ should follow a gamma function[32]. Figure 4c shows that the distribution of $t_{pause}$ is non-exponential at all ATP concentrations tested. Fitting the $t_{pause}$ distribution at 10 µM and 5 mM ATP to a gamma function indicates that this part of the remodelling reaction involves ~5 reaction steps (Fig. 4d).

Given the dependence of $t_{pause}$ on ATP concentration, we conclude that the pause period comprises ~5 ATP binding and hydrolysis events, which is approximately equivalent to the number of base pairs being translocated at the exit site. We also plotted $1/t_{pause}$ against ATP concentration and fitted a simple 1:1 binding isotherm (Fig. 4e), providing a pseudo-affinity of ATP for CHD4 of 4.3 ± 0.2 mM, within the range of estimated intracellular ATP concentration of 1–10 mM[33]. Together, these data show that

CHD4 is a processive remodeller that hydrolyses at least 5 ATP molecules per translocation step at the exit side.

**CHD4 binding causes significant dynamics at entry-side DNA.** Hitherto, we had only examined the behaviour of the exit-side DNA in the $0^{AF555}w60^{Bio}$ nucleosome. To probe the entry side behaviour, we constructed nucleosomes with the AF555 located 9 bp into the 60-bp extension on the entry side [$0w(9^{AF555})60^{Bio}$ nucleosomes, Fig. 5a]. In this case, most particles displayed FRET values of 0.2 or 0.4 (Supplementary Fig. 2a), reflecting more distantly located fluorophores. The two values most likely correspond to the presence of a distal and proximal AF647-labelled H2A (relative to the position of AF555 on the entry-side DNA), respectively. As we expected the FRET of $0w(9^{AF555})60^{Bio}$ to increase during remodelling, we used the distally labelled particles for entry-side analysis to maximize the resolution.

Concordant with our our observations for $0^{AF555}w60^{Bio}$, the $0w(9^{AF555})60^{Bio}$ nucleosome alone did not undergo significant FRET changes over time (Fig. 5b). In sharp contrast, addition of CHD4 in the absence of ATP resulted in significant time-dependent FRET changes. A gradual increase in FRET over the course of ~5–50 s gave rise to a long-lived FRET state at

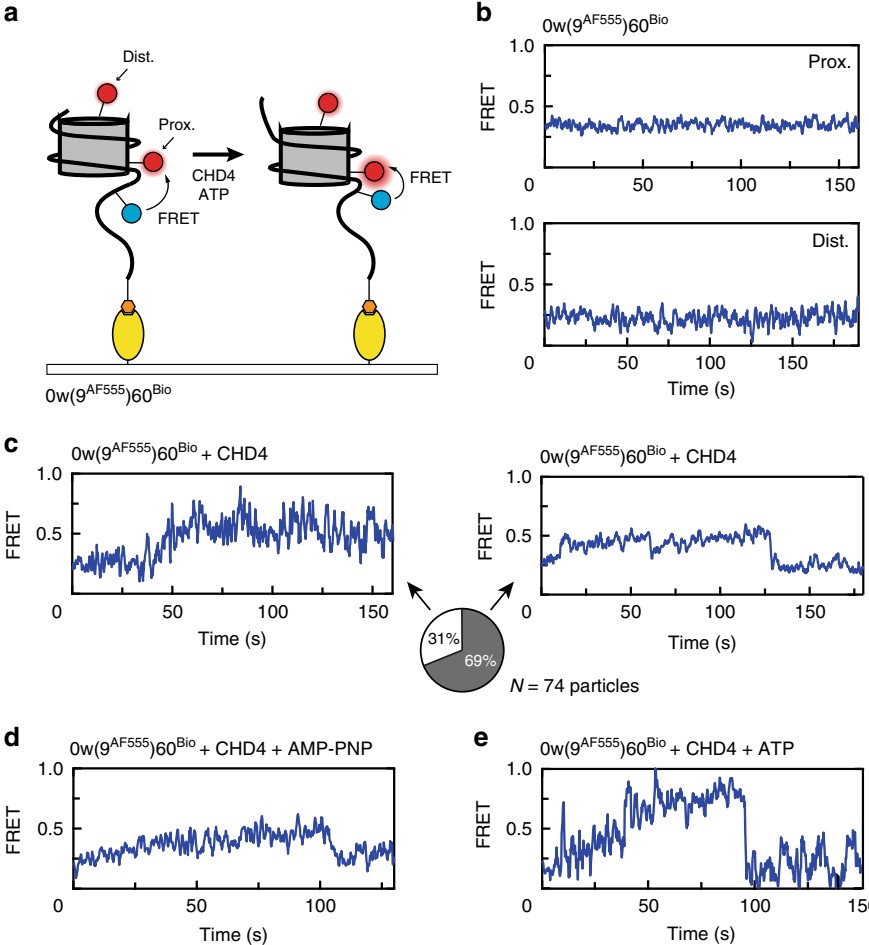

**Fig. 5 CHD4 binding induces changes in extra-nucleosomal DNA at the TA-poor side. a** Schematic showing the labelling scheme for $0w(9^{AF555})60^{Bio}$ and nucleosomal conformations before and after CHD4 remodelling. **b** FRET vs time traces of $0w(9^{AF555})60^{Bio}$ alone with a proximal (top) and distal (bottom) AF647-labelled H2A. **c** FRET vs time trace of $0w(9^{AF555})60^{Bio}$ bearing a distal AF647 label, showing a gradual increase upon the addition of 2 nM CHD4. This newly established structure can be relatively stable (left) or can be transient, dropping back to the initial state (right). The proportions of each scenario (from 74 molecules) are illustrated as a pie chart. **d** FRET vs time trace of $0w(9^{AF555})60^{Bio}$ bearing a distal AF647 label, showing an increase upon the addition of 2 nM CHD4 and 1 mM AMP-PNP. **e** FRET vs time traces of $0w(9^{AF555})60^{Bio}$ bearing a distal AF647 label showed a greater increase during CHD4 remodelling in presence of ATP, comparing to the changes induced by CHD4 alone.

0.45–0.5 (Fig. 5c), and we estimate that this change corresponds to ~1–3 bp movement of DNA. Interestingly, the FRET of most molecules (69%, 51 out of 74 particles) returned to its starting value, via a transition that was faster than the initial rise, whereas the remaining particles exhibited the higher FRET state until fluorophore bleaching occurred (Fig. 5c). Similar behaviour was observed in the presence of the ATP analogue AMP-PNP (Fig. 5d).

To observe entry-side behaviour during remodelling, we treated the $0w(9^{AF555})60^{Bio}$ nucleosomes with 2 nM CHD4 and 1 mM ATP. Under these conditions, distally labelled $0w(9^{AF555})60^{Bio}$ consistently displayed a gradual increase in FRET from 0.2 to ~0.6–0.8, significantly greater than the effect of CHD4 alone. Of the nucleosomes that underwent this increase, 64% subsequently exhibited a sudden return to ~0.2 (Fig. 5e). In some cases, this cycle was repeated over the observation timescale. The increase in FRET is consistent with a gradual movement of the entry-side DNA into the nucleosome in the presence of ATP, shortening the distance between the fluorophores. Nucleosomes bearing a proximally labelled H2A behaved similarly, with their FRET states increased from 0.4 to ~0.6–0.7 upon CHD4 binding (Supplementary Fig. 2b) and a greater increase to ~0.8 in presence of CHD4 and ATP (Supplementary Fig. 2c), both followed by a return to ~0.4.

We interpret the sharp drop in FRET observed as 'reversions' in which the DNA returns to its starting position after unsuccessful remodelling. Gel-based remodelling assays show that this entry-side labelled nucleosome cannot be remodelled to the same extent as unlabelled or exit-side labelled nucleosomes (Supplementary Fig. 3), perhaps due to steric hindrance caused by the bulky fluorophore on the DNA.

These data show that the binding of CHD4—either alone or in the presence of AMP-PNP—induces dynamic changes in nucleosome structure at the entry side and that ATP-driven remodelling proceeds with a gradual increase in FRET at the entry side alongside discrete downward jumps at the exit side.

**CHD4 binding induces dynamics at both nucleosomal DNA ends.** The observations above suggest that CHD4 binding causes a change in extra-nucleosomal DNA conformation on the entry side (or TA-poor side; Fig. 1b) that decreases the distance between the two fluorophores. One possible explanation is that DNA at the entry side translocates a short distance into the nucleosome even upon remodeller binding in the absence of nucleotide. A similar effect has been observed for CHD1; remodeller binding caused a 1–3 bp shift of the DNA in the TA-poor region[19].

Because our gel-based data indicate that the nucleosome has two CHD4 binding sites and that CHD4 can remodel both 0w60 and 60w0 substrates, we hypothesized that remodelling can occur in either direction, depending on the site of CHD4 binding. Thus, either end of the nucleosome can act as an entry side providing it has sufficient extra-nucleosomal DNA. To test this idea, we assembled $9^{AF555}w60^{Bio}$ nucleosomes (Fig. 6a), which, again, displayed no significant dynamic behaviour alone (Fig. 6b). However, the addition of CHD4 caused an increase in FRET from 0.2 to ~0.4 for distally labelled nucleosomes (Fig. 6c), and 0.4 to ~0.6 for their proximal labelled counterparts (Supplementary Fig. 2d). Out of 78 distally labelled molecules, 34 were observed to return to the initial FRET state, whereas 44 remained at the higher FRET value. These data show that CHD4 elicits similar changes to flanking DNA at both the TA-rich and TA-poor ends of the 601-based nucleosome.

**The '9' end of 9w60 can only be remodelled as an exit side.** We also treated $9^{AF555}w60^{Bio}$ nucleosomes with 2 nM CHD4 and 1 mM ATP. Surprisingly, we observed similar FRET changes as were observed with the addition of CHD4 alone; that is, increases in FRET from 0.2 to ~0.4, followed often by a return to 0.2, are observed for distally H2A-labelled particles (Fig. 6d). These results suggest that processive remodelling in which the '9' end of the DNA acts as an entry site is unable to proceed, possibly because a 9-bp flanking sequence is too short to be translocated without destabilizing histone–DNA contacts. In contrast, 41% of the particles showed clear stepwise drops in FRET, indicating that $9^{AF555}w60^{Bio}$ nucleosomes are indeed able to be remodelled—but in the opposite direction—with the '9' end acting as an exit side (Fig. 6e). Our gel-based assays using $9^{AF555}w60$ as a substrate support this interpretation: that this nucleosome can only be remodelled to a more symmetrical configuration (Supplementary Fig. 3), consistent with the conclusion drawn from Fig. 1e (that CHD4 does not efficiently remodel shorter flanking DNA sequences towards the nucleosome). The roughly equal (65:45) split between particles that display initial increases versus decreases in FRET is consistent with the idea that remodelling can occur in either direction, most likely through CHD4 binding at each of its two possible binding sites.

Overall, these data demonstrate that nucleosome binding by CHD4 induces significant slow-timescale changes into extra-nucleosomal DNA. The data also point towards a binding mode for CHD4 that is distinct from that of CHD1[21], in that there is no significant displacement of several turns of DNA away from the nucleosome surface upon CHD4 binding.

## Discussion

Prior to this work, little was known about the CHD4-driven nucleosome remodelling mechanism. In contrast, a substantial body of work exists on yCHD1 structure and function. Human CHD4 and yCHD1 ATPase domains share 65% sequence similarity and 12 of the 19 residues in the yCHD1 ATPase that contact DNA are identical in hCHD4 (Supplementary Fig. 4). Furthermore, structure prediction algorithms predict that the fold of the CHD4 translocase domain closely resembles that of yCHD1[34].

EMSA data for CHD1 and CHD4 are also comparable. Ryan et al. estimated the dissociation constant for a complex formed between CHD1 and a 0w47 nucleosome to be around 30 nM[35], close to our observed value of 40–80 nM for CHD4. In addition, CHD1 also causes a second band shift at higher concentrations, which later was confirmed to represent a 2:1 complex in which each SHL2 site was occupied by a CHD1 molecule[36,37]. The correspondence of the yCHD1 gel-shift data with ours[35] indicates that CHD4 probably also engages nucleosomes at the SHL+2 and SHL−2 sites.

Regulation of remodelling activity is likely to be distinct for the two enzymes, however; CHD1 has a C-terminal DNA-binding domain that lifts two turns of DNA from the histone octamer surface at SHL7 of the other gyre[36]. This domain is absent from CHD4 and the deletion of the C-terminal third of CHD4 (which contains domains of unknown structure that mediate interactions with other proteins[38,39]) does not reduce its chromatin occupancy[7]. In addition, we observed that CHD4 remodels symmetric substrates such as 30w30 more completely than asymmetric substrates (e.g., 0w60), whereas CHD1 behaves in the opposite manner[28,40]. Interestingly, less CHD1 is required to remodel symmetric substrates if the DNA-binding domain is removed[40], suggesting that this domain might act as a DNA length sensor that reduces the ability of CHD1 to remodel the nucleosome to the end of a DNA sequence. Our data show that CHD4 does not possess this regulatory function.

We used smFRET to monitor distance changes between the DNA and the histone octamer during ATP-dependent

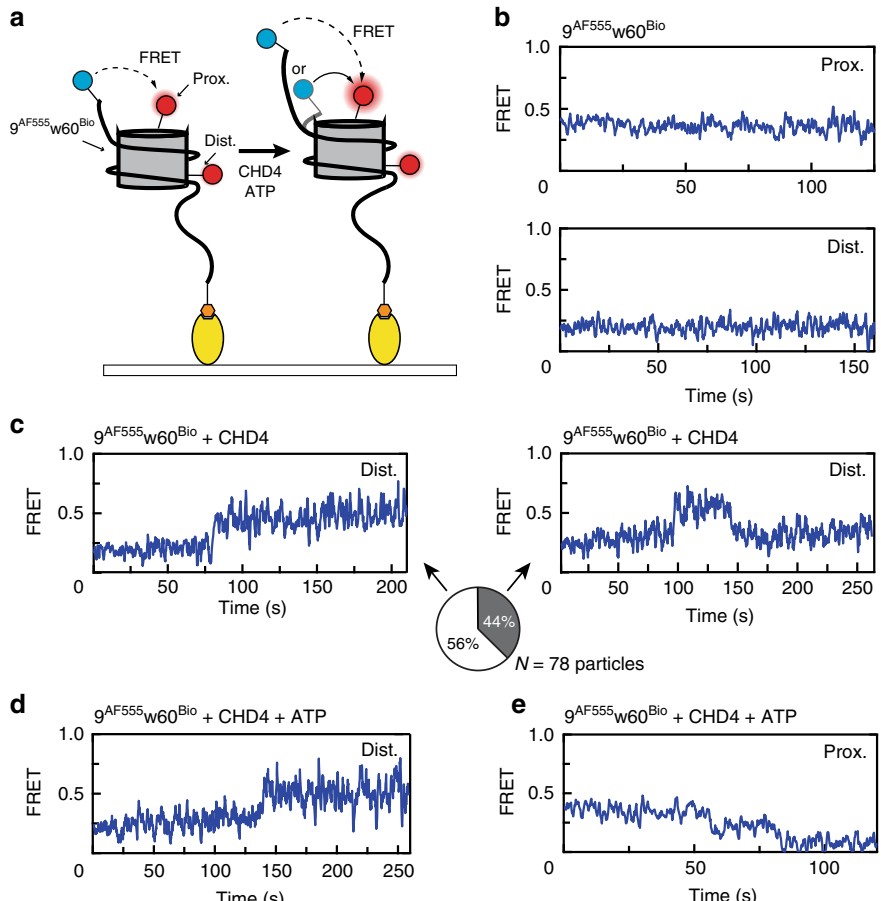

**Fig. 6 CHD4 induces similar FRET changes in extra-nucleosomal DNA at the TA-rich side. a** Schematic showing the labelling scheme for $9^{AF555}w60^{Bio}$ and the two possible remodelling directions after treating with CHD4, depending on whether the '9' end acts as an entry or an exit side. **b** FRET vs time traces for $9^{AF555}w60^{Bio}$ alone with a proximal (top) or distal (bottom) H2A AF647 label. **c** FRET vs time trace for $9^{AF555}w60^{Bio}$ bearing a distal AF647 label, showing an increase upon the addition of 2 nM CHD4. This newly established structure can be relatively stable (left) or can be transient, dropping back to the initial state (right). The proportions of each scenario (from 78 particles) are illustrated as a pie chart. **d** FRET vs time trace for distally labelled $9^{AF555}w60^{Bio}$ in presence of both 2 nM CHD4 and 1 mM ATP, showing an increase to a similar level to that of CHD4 alone. **e** FRET vs time trace for proximally labelled $9^{AF555}w60^{Bio}$, showing a stepwise drop upon treatment with 2 nM CHD4 and 1 mM ATP. In this case, the AF555 tag at the '9' end is moving away from the histone octamer, in contrast to the direction of movement observed in (**d**).

nucleosome remodelling. When focusing on the DNA exit end, we observed FRET changes representing 4–6 bp of DNA translocation away from the histone octamer; these changes were consistent across a wide range of ATP concentrations. Related observations have been made for the ISWI, RSC and CHD1[13–15]. In the case of ISWI, it was shown that each multiple-base-pair movement comprised a cluster of 1-bp steps that could be separated by lowering the ATP concentration. In contrast, however, a reduction in ATP concentration during CHD4 remodelling results only in an increase in the length of $t_{pause}$; CHD1 has been reported to behave similarly[14]. These data indicate that the expulsion of 4–6 bp from the exit side is a concerted process.

Remodelling rate (defined as the length of $t_{pause}$) was insensitive to CHD4 concentration, but highly dependent on ATP concentration, indicating a processive mode of CHD4 remodelling and that ATP turnover is the rate-limiting step. We also observed that the $t_{pause}$ distribution for CHD4 was essentially a constant shape across a 2000-fold range of ATP concentrations. This distribution implies that the pause time comprises multiple rate-limiting intermediate steps, and a fit of the distribution indicates that a reaction sequence of more than three steps best explains the data. That is, the data point to a model in which four or more ATP molecules are consumed during $t_{pause}$, prior to the

concerted exit-side translocation of 4–6 bp of DNA. Considering this inference together with our estimates of DNA translocation distance, we propose that that each base-pair movement might require ~1 ATP hydrolysis event, which is in the range of mechano-chemical conversion efficiencies measured for other DNA helicases such as RecQ and PcrA, as well as for CHD1[36,41,42].

We unexpectedly observed that the binding of CHD4 alone was sufficient to trigger significant conformational changes in the flanking entry-side DNA. These changes were observed for both $9^{AF555}w60^{Bio}$ and $0w(9^{AF555})60^{Bio}$ nucleosomes, indicating that the effect is independent of whether the flanking DNA is on the TA-rich or TA-poor side of the 601 sequence. Furthermore, these changes were gradual—a rise of ~0.2 FRET units over 5–50 s, followed in many cases by a more abrupt decrease to the original value. We interpret these changes as a reversible movement of flanking DNA towards the histone octamer, the energy for which is provided simply by binding of the remodeller. Following previously described models[17,18], we propose that this movement occurs via a corkscrew-like twisting of the DNA—perhaps even of just one strand, to best maintain histone–DNA interactions. No such changes were observed for $0^{AF555}w60$ nucleosomes in the presence of CHD4, supporting the idea that the movement of the

'0' end into the nucleosome core would be too unfavourable because of the loss of histone–DNA interactions.

These data are consistent with the recent structure of a Snf2-nucleosome complex in the absence of nucleotide[43]. In this structure, the Snf2 induces a 1-bp translocation in one strand (the tracking strand, which orients $5' \rightarrow 3'$ toward the dyad[44]) of the DNA at the SHL2 site to which the remodeller is bound. This effect appears to be propagated as far as the entry side[43,45] and could very possibly extend into the flanking DNA. Analysis of histone–DNA crosslinking in CHD1-nucleosome complexes also supports this idea, with the observation of a twist/shift of 1–3 bp at SHL5 upon CHD1 binding[19]. Indeed, distortions at the SHL2 site have been observed in structures of the nucleosome in isolation[46–48], suggesting that this location might be predisposed to be an initiation point for remodelling.

We observed comparable changes in entry-side FRET when incubating CHD4 and nucleosome in either the absence or presence of the non-hydrolysable nucleotide analogue AMP-PNP. This observation corroborates structural and smFRET data for Snf2[43] and also CHD1 crosslinking data[19]. For both remodellers, the DNA distorting effect was stronger with apo or AMP-PNP-bound remodeller. In this case the two lobes of the translocase domain are in a so-called 'open' conformation relative to each other[45,49]. In contrast, these changes in DNA structure are not observed for CHD1 or Snf2 in the presence of ADP·BeF_3, an ATP analogue that induces a more 'closed' conformation involving a rotation of one lobe[36,43,50]. Because distortion of the DNA tracking strand appears to be a general feature of SF2-family helicases bound in the open/apo state[51], and our AF555 fluorophores were attached on the extra-nucleosomal part of the tracking strand, our data suggest that CHD4 behaves in a similar manner as Snf2. That is, binding of CHD4 draws the tracking strand in from the entry side. An important aspect of the process that we report here is its slow timescale (up to tens of seconds). The reason for this timescale is not currently clear; it is possible that the required changes are coupled to slow, large-scale fluctuations of nucleosome structure that are gradually 'captured' by CHD4.

In the presence of both CHD4 and ATP, the FRET measured for $0w(9^{AF555})60^{Bio}$ increases from 0.2 to ~0.8, indicating that DNA enters the nucleosome in a steady manner until the AF555 fluorophore prevents further translocation and the DNA is 'reset' to its starting position. This gradual increase contrasts sharply with the concerted ~5-bp translocations that we observe at the exit side and, taken together with the other available data, suggests a mechanism for CHD4 remodelling in which entry side and exit side movements are decoupled from each other.

We propose that CHD4 first binds SHL2 in an open conformation, inducing a 1-bp shift in the tracking strand that is propagated from SHL2 back to the entry side (Fig. 7a, b). This change is reversible, perhaps through conformational changes or CHD4 dissociation. Second, binding of ATP closes the two ATPase lobes, realigns the tracking and guide strands, and induces a distortion of the DNA between the SHL2 site and the dyad (Fig. 7c). This process 'relaxes' the DNA between SHL2 and the entry side. Third, following ATP hydrolysis, the two lobes of CHD4 return to the open position, establishing a new 1-bp translocation of the tracking strand that pulls a nucleotide from the entry side (Fig. 7d). This is consistent with the available structural and biochemical data that show an open position for SF2-family remodellers in the ADP-bound states[43]. This cycle continues until a ~5-bp distortion is built up near the dyad (Fig. 7e). The nature of this strained state is not yet known, but it could either be a small loop or a small segment of overwound DNA helix[6]. Eventually, the strain becomes too great to be maintained and is released at the exit side by a concerted twist of

the DNA (Fig. 7f), moving DNA relative to the histone octamer. This entire process is animated in Supplementary Movie 1.

In conclusion, we have carried out the first mechanistic analysis of CHD4-driven chromatin remodelling and our data lead to a model in which processive CHD4 action builds up strain in the DNA through 4–6 successive ATP-dependent translocations of 1 bp, and the strain is released by a concerted expulsion of those base pairs from the exit side. This model synthesizes a wide range of recent structural and mechanistic measurements on several other chromatin remodelling enzymes, suggesting common mechanistic features across SF2-family remodellers, while at the same time indicating that some differences exist in the precise mechanisms of action of each remodeller.

## Methods

**Histone purification and labelling.** Recombinant human histones were expressed in *Escherichia coli* (*E. coli*) BL21 (DE3) pLysS cells and purified from inclusion bodies. Competent cells were transformed with histone expression plasmids (pET28a, Novagen) and grown in LB medium at 37 °C until reaching an OD600 of 0.6. Protein expression was induced by the addition of 1 mM IPTG for 4 h at 37 °C. Cell pellets were harvested and lysed by sonication in histone lysis buffer containing 50 mM Tris-HCl pH 7.5, 100 mM NaCl, 1 mM beta-mercaptoethanol (BME), and the insoluble fraction after clarification (30 min at 15,000 × g at room temperature) was washed twice with histone lysis buffer containing 1% (v/v) Triton X-100, and then twice without Triton X-100. The pellet was dissolved in 10 mL of unfolding buffer (20 mM Tris-HCl pH 7.5, 6 M guanidinium HCl and 1 mM dithiothreitol (DTT)) per L of culture by stirring at room temperature overnight. Resuspended pellets were then centrifuged at 20,000 × g for 30 min at 4 °C. Filtered supernatants were injected onto a preparative Vydac protein and peptide C18 column (300 Å pore size, Catalogue No. 218TP1022) at a flow rate of 7 mL/min (20–70% acetonitrile over 40 min, 0.1% TFA). The fractions containing the target protein (as judged by UPLC-MS analysis) were freeze-dried and stored at –20 °C in sealed containers.

To generate H2A site-specifically labelled with AF647, a T120C mutant was generated by site-directed mutagenesis. This construct was purified by reverse-phase HPLC using the same method employed for the wild-type proteins. After lyophilization, 200 nmol of H2A T120C was dissolved in 0.5 mL of unfolding buffer containing 1 mM tris(2-carboxyethyl)phosphine (TCEP), followed by degassing. The solution was then mixed directly with a 5× molar excess of Alexa Fluor™ 647 C_2 maleimide (Thermo Fisher Scientific A20347), and then incubated at room temperature for 10 min and then at 4 °C overnight. The reactions were quenched via the addition of 30 mM BME and then purified via gel filtration on a Superdex 200 10/300 column in 20 mM Tris, pH 7.0, 7 M guanidine HCl, 0.1% (v/v) BME. Purified labelled H2A was dialysed against deionized water with 0.05% (v/v) BME overnight at 4 °C and lyophilized for long-term storage. Labelling efficiency was ~50%, determined using method described in ref. [52].

**Nucleosomal DNA preparation.** DNA oligonucleotides were made by PCR from a plasmid containing a 601 positioning sequence[22]. The PCR primers contained 5′ AF555 or biotin-TEG modifications (Integrated DNA Technologies, Singapore) to install these modifications at the indicated locations. The PCR products were first concentrated by ethanol precipitation and were redissolved in 1× TE buffer (10 mM Tris-HCl pH 8, 1 mM EDTA), and then cleaned up using phenol:chloroform: isoamyl alcohol (25:24:1) extraction, followed by a chloroform wash to remove residue phenol. After isopropanol precipitation and washing with 70% (v/v) ethanol, the DNA pellet was dissolved in 1× TE and separated on a 0.5× TBE 5% polyacrylamide gel. The DNA band with the desired size was cut from the gel and electroeluted into 0.5× TBS at room temperature. The final DNA product was concentrated by ethanol precipitation overnight and the resulting pellet was dissolved in 1× TE.

**Nucleosome preparation.** Histone octamer was reconstituted as described in ref. [46]. Mononucleosomes were assembled by salt gradient dialysis using a double dialysis method[53]. After mixing labelled octamer and DNA at a 1:0.95 molar ratio in 10 mM Tris-HCl pH 7.5, 2 M NaCl, 1 mM EDTA, the mixture was loaded into a small dialysis button, which was then placed into a dialysis bag containing 30 mL of 10 mM Tri-HCl pH 7.5, 2 M NaCl, 1 mM EDTA and 0.1 mM DTT. The bag was then dialysed against 2 L of 1× TE containing 0.1 mM DTT overnight at room temperature.

The next day, the dialysis button was dialysed further against 10 mM Tris pH 7.5, 2.5 mM NaCl and 0.1 mM DTT. Content in the dialysis button was harvested and the nucleosome quality was checked by 0.5× TBE 5% polyacrylamide gel electrophoresis at 150 V in the cold room.

**Expression and purification of FLAG-CHD4 in HEK293 cells.** Suspension-adapted HEK Expi293F™ cells (Thermo Fisher Scientific, Waltham, MA, USA) were

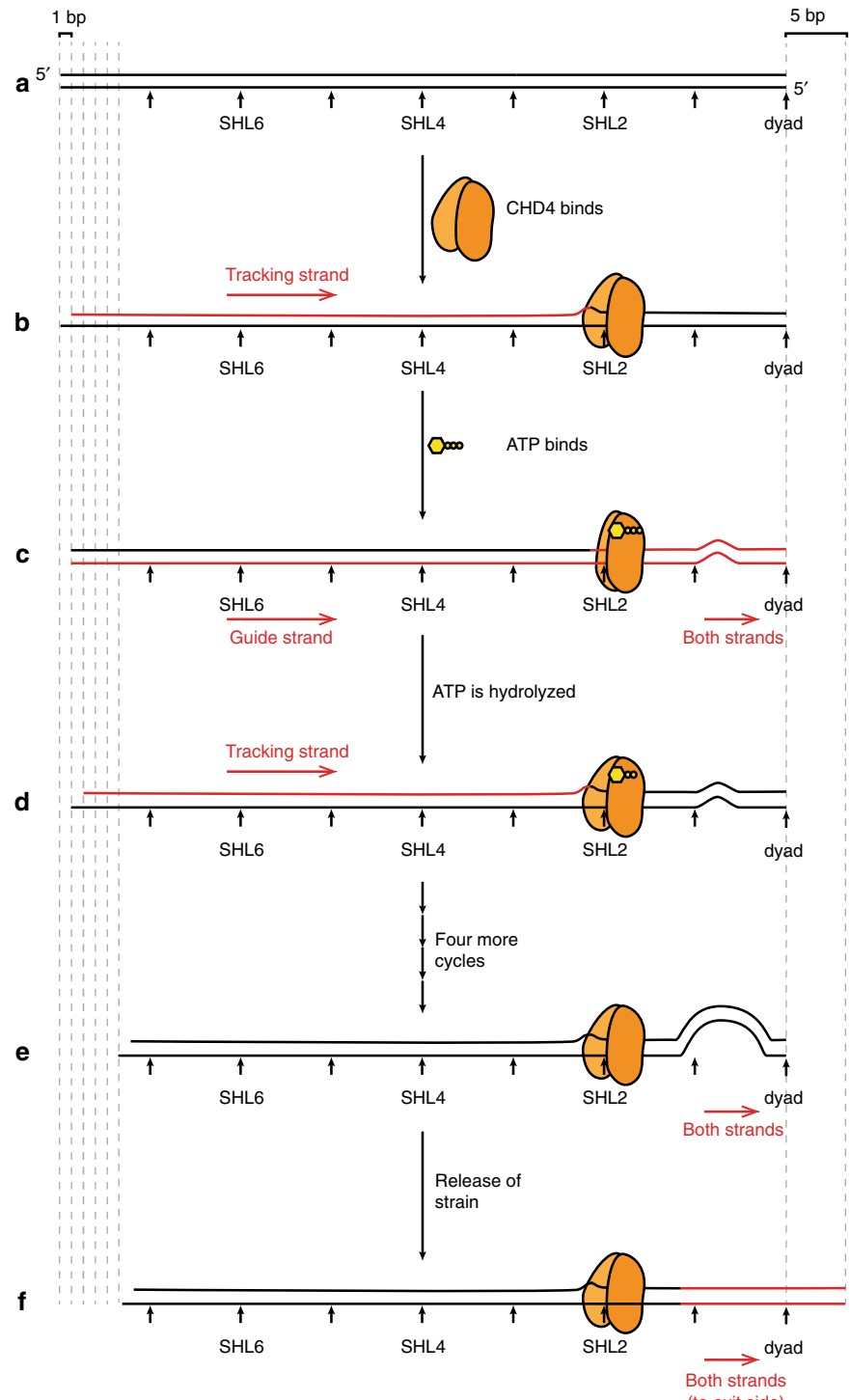

**Fig. 7 A model for CHD4-driven nucleosome sliding. a** Schematic of one-half of a nucleosomal DNA sequence. SHL positions are indicated. **b** Binding of the two-lobed ATPase domain of CHD4 (not drawn in scale) induces a 1-bp shift in the so-called tracking strand of the DNA, creating a distortion that reaches from the SHL2 site all the way back to the 5′ end of the tracking strand. The DNA segment moved by CHD4-induced remodelling is coloured in red in this and the following figures. **c** Binding of ATP induces a conformational change in CHD4, 'closing' the two lobes around the ATP. This change results in the guide strand 'catching up' to the tracking strand and then the movement of both strands by ~1 bp past CHD4, so that a small bulge (or other irregularity) forms in the region of the dyad. **d** Hydrolysis of ATP drives a return of CHD4 to the open conformation, inducing a second 1-bp movement of the tracking strand, analogous to that in part **b**, and initiating a second cycle of the same process. **e** Following four further cycles of ATP binding and hydrolysis, a large irregularity is built up near the dyad. **f** The strain induced by this irregularity causes a concerted rearrangement of the DNA such that 5 bp are expelled from the exit side. This whole cycle can in principle be repeated many times.

grown to a density of $2 \times 10^6$ cells/mL in Expi293™ Expression Medium (Thermo Fisher Scientific). pcDNA3.1 plasmids encoding for FLAG-CHD4 were transfected into cells using linear 25-kDa polyethylenimine (PEI, Polysciences, Warrington, PA, USA). Fifty micrograms of the DNA mixture was first diluted in 2.7 mL of PBS and vortexed briefly. One hundred micrograms of PEI was then added, and the mixture was vortexed again, incubated for 20 min at room temperature, and then added to 25 mL of HEK cell culture. The cells were incubated for 65 h at 37 °C with 5% $CO_2$ and horizontal orbital shaking at 130 rpm.

Cells were harvested and washed twice with PBS, centrifuged ($300 \times g$, 5 min), snap-frozen in liquid nitrogen and stored at –80 °C. Lysates were prepared by sonicating thawed cell pellets in 0.5 mL of lysis buffer [50 mM Tris/HCl, 500 mM NaCl, 1% (v/v) Triton X-100, 1× cOmplete EDTA-free protease inhibitor (Roche, Basel, Switzerland), 100 mM ATP, 0.2 mM DTT, pH 7.9], and then clarifying the lysate via centrifugation ($\geq 16{,}000 \times g$, 20 min, 4 °C). The cleared supernatant was used for FLAG-affinity pulldowns.

Two hundred microlitres of anti-FLAG Sepharose 4B beads [Biotool, Houston, TX, USA; pre-equilibrated with 50 mM Tris-HCl, 150 mM NaCl, 0.1% (v/ v) Triton X-100, 1× cOmplete EDTA-free protease inhibitor, 0.2 mM DTT, pH 7.5] was added to 15 mL of cleared HEK cell lysate. The mixtures were incubated overnight at 4 °C with orbital rotation. Post-incubation, the beads were washed with 5× wash buffer [50 mM Tris-HCl, 150 mM NaCl, 0.5% (v/v) IGEPAL CA630, 100 mM ATP, 0.2 mM DTT, pH 7.5]. Bound proteins were eluted by 5× 400 μL treatment with 'elution' buffer (10 mM HEPES, 150 mM NaCl, 150 μg/mL 3× FLAG peptide (MDYKDHDGDYKDHDIDYKDDDDK), pH 7.5) for 1 h at 4 °C. Concentrations of CHD4 in the elution fractions were estimated using densitometry (ImageJ) by loading onto a SDS-PAGE along with a known amount of BSA and staining with SYPRO® Ruby.

**NuRD purification**. NuRD was purified as described in ref. [54]. Briefly, GST-FOG1 (1–45) was overexpressed in *E. coli* BL21(DE3) cells. The cells were lysed via sonication in GST binding buffer (50 mM Tris, 150 mM NaCl, 0.1% BME, 0.5 mM PMSF, 0.1 mg/ml lysozyme, 10 μg/ml DNase I, pH 7.5) and clarified via centrifugation ($\geq 16{,}000 \times g$, 20 min, 4 °C). The cleared supernatant was then incubated with pre-equilibrated glutathione-Sepharose 4B beads (GE Healthcare) for 1 h at 4 °C. The beads were then washed in GST wash buffer (20 CV, 50 mM Tris, 500 mM NaCl, 1% (v/v) Triton X-100, 1 mM DTT, pH 7.5) and then NuRD binding buffer (10 CV, 50 mM HEPES-KOH, 150 mM NaCl, 1% v/v Triton X-100, 1 mM DTT, 1× cOmplete® protease inhibitor (Roche), pH 7.4). The beads were then used for NuRD pull-down experiments.

MEL cell nuclear extracts were prepared by incubating the thawed cell pellets with hypotonic lysis buffer (5 ml/g of cells; 10 mM HEPES-KOH, 1.5 mM MgCl₂, 10 mM KCl, 1 mM DTT, cOmplete® protease inhibitor, pH 7.9) for 20 min at 4 °C. IGEPAL® CA-630 was then added (final concentration, 0.6% v/v), and the cells were then further incubated for 10 min. The mixture was then vortexed for 10 s and then centrifuged ($3300 \times g$, 5 min). The cytoplasmic supernatant was discarded, and the nuclear pellet was gently washed once with lysis buffer (+0.6% (v/v) IGEPAL® CA-630). The washed nuclear pellet was resuspended in NuRD binding buffer (3 mL/g of cells), then lysed by sonication, and incubated on ice for 30 min to allow the chromatin to precipitate. The nuclear extract was then clarified via centrifugation ($\geq 16{,}000 \times g$, 20 min, 4 °C), and the cleared supernatant was incubated with Streptavidin beads (a preclearing step for FOG1(1–45) peptide affinity purification) before incubating with the above FOG1 affinity resins overnight at 4 °C. Post-incubation, the nuclear extract was then washed with 20 CV of NuRD wash buffer 1 (50 mM HEPES-KOH, 500 mM NaCl, 1% (v/v) Triton X-100, 1 mM DTT, pH 7.4) and then 10 CV of NuRD wash buffer 2 (50 mM HEPES-KOH, 150 mM NaCl, 0.1%(v/v) Triton X-100, 1 mM DTT, pH 7.4). Captured proteins were eluted with GST-FOG1(1–45) elution buffer (50 mM reduced glutathione, 50 mM HEPES-KOH, 150 mM NaCl, 0.1% Triton X-100, 1 mM DTT, pH 8.0) for 30 min at 4 °C. This elution step was repeated at least twice to ensure complete elution, and the concentration was estimated using densitometry as described above.

**Nucleosome repositioning assay**. Reactions contained 50 nM of either labelled or non-labelled nucleosomes, 50 mM Tris pH 7.5, 50 mM NaCl, 3 mM MgCl₂, and the enzyme concentrations varied from 0 to 10 nM. The reactions were incubated at 37 °C (unless otherwise indicated) and then stopped by placing them on ice and the addition of 0.5 μg salmon sperm DNA or competitor DNA and 4% (w/v) sucrose prior to electrophoresis on 0.5× TBE 5% polyacrylamide gels. Gels were stained with 1× SYPRO® Gold and then imaged by on an FLA-9000 laser scanner.

**Nucleosome pull-down assay**. HEK Expi293F™ culture expressing FLAG-CHD4 (3 mL) was prepared as described above. The cell pellet was then lysed in the same way and loaded onto anti-FLAG beads. After five washes, the beads were washed three times again with 10 mM Tris pH 7.5, 2.5 mM NaCl and 0.1 mM DTT, and split into 3 aliquots, and each was incubated with the same amount of nucleosome (~3 pmol in 10 mM Tris pH 7.5, 2.5 mM NaCl and 0.1 mM DTT) at 4 °C overnight. The next day, the proteins were eluted using the method as above and equivalent amounts of input and elution were checked by SDS-PAGE.

**EMSA of nucleosome–CHD4 interaction**. Each reaction contained 60 nM of AF647-labelled nucleosomes, 50 mM Tris pH 7.5, 50 mM NaCl, 3 mM MgCl₂, and the enzyme concentrations as indicated in the figures. The reactions were incubated on ice for 60 min, protected from light, and then mixed with 4% (w/v) sucrose prior to electrophoresis on 0.5× TBE 5% polyacrylamide gels. Gels were then imaged on an FLA-9000 laser scanner.

**Single-molecule instrument setup**. An Olympus IX-71 based model was modified to build an objective type total internal reflection fluorescence (TIRF) microscope to record single-molecule movies. A coherent Sapphire green (532 nm) laser was used to excite donor (AF555) molecules by focusing onto a ×100 oil immersed objective and scattered light was removed using a 560-nm long pass filter. Donor and acceptor (AF647) signals were collected at 565 and 665 nm using a band pass filter (560–600 nm and a long pass filter at 650 nm, respectively). Then, both signals were first split by a 638-nm dichroic mirror using Photometrics Dual View (DV-2) and then were focused onto a CCD camera (Hamamatsu C9 100-13), simultaneously. Single-molecule movies were recorded at 5 frames per second.

**Preparation of PEGylated coverslips**. First, quartz coverslips were sonicated with 2–5 M KOH for 20 min and rinsed with double distilled water (ddH₂O). Second, aminosilation of coverslips were carried out in a mixture of 100 mL water and 1% (v/v) aminopropylsilane (Alfa Aesar, A10668, UK). Third, PEGylation was carried out by incubating a mixture of biotinPEG-SVA and mPEG-SVA (Laysan Bio, AL, USA) in the ratio of 1:20 prepared in 50 mM MOPS at pH 7.5 on the top of the silanized coverslip for 3–4 h. Finally, PEGylated coverslips were rinsed with ddH₂O, dried with dry nitrogen and stored under dry nitrogen gas at –20 °C.

**Single-molecule FRET experiments**. Immuno-pure neutravidin solution was prepared in imaging buffer (40 mM Tris-HCl, pH 7.5, 12 mM HEPES, pH 7.5, 3 mM MgCl₂ and 60 mM KCl, 0.32 mM EDTA, 10% (v/v) glycerol, 0.02% (v/v) IGEPAL (Sigma-Aldrich) and spread on the top of dry PEGylated coverslip for 10 min. Then, polydimethylsiloxane (PDMS) was sandwiched on the top of the neutravidin coated coverslip to create a microscopic channel. Then, blocking buffer (prepared by mixing 1% (v/v) Tween-20 in imaging buffer) was injected onto the microscopic channel, in order to reduce non-specific binding of proteins on the surface, and incubated for 10–15 min.

Different mononucleosome samples labelled with FRET pair fluorophores and biotin were diluted to 50 pM in imaging buffer and injected into the flow chamber using a syringe pump (ProSense B.V.); the mixture was then incubated for 5–10 min. Unbound sample was removed by flowing imaging buffer through the chamber. Next, an oxygen-scavenging system (OSS) consisting of protocatechuic acid (PCA, 2.5 mM) and protocatechuate-3,4-dioxygenase (PCD, 50 nM) in imaging buffer were flowed across the surface to reduce photobleaching of the fluorophores. Trolox (2 mM) was also added to reduce photoblinking of dyes. Multiple movies were then recorded to measure the distribution of free nucleosomes. For CHD4 binding assays, a mixture of CHD4 (2–20 nM) diluted in imaging buffer (containing OSS) was injected for 10–20 s while a movie was recorded continuously. For remodelling assays, additional ATP (0.01–20 mM) was included in the imaging buffer. The CHD4 binding or remodelling mixture reached the reaction chamber in 10–20 s and a movie was recorded continuously at room temperature (20 ± 1 °C) for 3–5 min (until acceptor dyes photobleached).

**Data analysis**. Single-molecule intensity time trajectories were generated in interactive data language (IDL) and these trajectories were analysed in MATLAB using home written scripts (https://cplc.illinois.edu/software/). An approximate FRET value is measured as the ratio of acceptor intensity to the sum of the donor and acceptor intensities after correcting for cross-talk between donor and acceptor channels.

Since the acceptor dye was on the H2A subunit of the histone octamer, it gives rise to heterogeneity in FRET population, because three different labelling states exist: (i) acceptor dye proximal to donor dye, yielding a mid-FRET state; (ii) acceptor dye distal to donor dye, giving a low-FRET state; (iii) two acceptor dyes on both H2A subunits, yielding a high-FRET state. For example, we observed a distribution that could be modelled by three Gaussians with mean FRET values of 0.78, 0.67 and 0.42 for exit side 0w60 nucleosomes. Similarly, for entry side labelled nucleosome (0W9AF555-60) we observed peaks with means of 0.62, 0.38 and 0.23. In this study, we focused on the molecules in the mid-FRET state. In the FRET trace analysis, we chose molecules showing a mean FRET value in the range of 0.55–0.7 and displaying a single acceptor photobleaching step. In some cases, we also selected populations showing mean FRET values in the range of 0.35–0.5 and showing a single step acceptor photobleaching, in order to analyze distal side dynamics. These selection criteria allowed us to distinguish between nucleosomes bearing one H2A labelled subunit. Similar criteria with different FRET cut-off values were applied to select mid-FRET traces in different nucleosomes ($n = 3$, 6 and 9) for calibration purpose. For entry site labelled nucleosomes, distally labelled mononucleosomes were chosen for statistical analysis.

To measure the rate of remodelling reaction at different ATP or CHD4 concentrations, histograms were generated in each condition and fit with a

Gaussian, where the peak of the curve fit was taken as the mean time for remodelling by CHD4.

**Gamma distribution**. To measure the number of steps involved in a remodelling reaction, a Gamma distribution was applied to time-binned histograms of the form:

$$f(t) = \frac{k^N t^{N-1}}{\Gamma(N)} \exp(-kt) \tag{1}$$

where $k$ is the rate of the reaction and $N$ is the number of steps hidden in the reaction.

**Reporting summary**. Further information on research design is available in the Nature Research Reporting Summary linked to this article.

## Data availability

Data supporting the findings of this paper are available from the corresponding authors upon reasonable request. A reporting summary for this Article is available as a Supplementary Information file. The source data underlying Figures 1C–F, 3B–F, 4A–E, 5B–E, 6B–E, Supplementary Fig. 1A–G, and Supplementary Fig. 2A–D and 3 are provided as a Source Data file.

## Code availability

Custom code scripts used for data analysis in IDL and MATLAB program are publicly available at https://cplc.illinois.edu/software/.

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

## Acknowledgements

The work was funded by the following grants from the National Health and Medical Research Council of Australia: APP1012161, APP1063301, APP1126357 and a fellowship from the same organization to J.P.M. (APP1058916). A.M.O. is an Australian Research Council Laureate Fellow.

## Author contributions

Conceptualization: Y.Z., B.P.P., A.M.O. and J.P.M.; methodology: Y.Z., B.P.P., D.P.R., J.K.K.L., C.F., K.P. and M.J.B.; software: B.P.P.; investigation: Y.Z. and B.P.P.; resources: R.J.P., A.M.O. and J.P.M.; writing—original draft: Y.Z. and J.P.M.; writing—reviewing and editing: Y.Z., B.P.P., M.J.B., R.J.P., A.M.O. and J.P.M.; supervision: A.M.O. and J.P.M.

## Competing interests

The authors declare no competing interests.
