## [Peer Review File · Nature Communications]

Reviewers' comments:

Reviewer #1 (Remarks to the Author):

Zhong and Paudel and colleagues present in this manuscript a study of the nucleosome remodeler hCHD4. In gel based assays, they observed that CHD4 has no preferred remodeling direction, except that it tends to symmetrize free DNA around the histone-octamer. They then transferred their assay to the single-molecule level and used FRET to probe the entry and exit site dynamics of CHD4-nucleosome complexes. They find that DNA exits in large steps of 4 to 6 bp, while the entry appeared slow and gradual. Based on their data they propose a model of CHD4 activity.

While this is an exciting study of a highly relevant protein-nucleosome interaction, at the current stage I do have several open questions and comments, which should be answered before publication:

Major:

- The introduction of this manuscript appears cluttered at the moment. After reading it several times, I am still not sure if I can follow it. In the discussion the authors compared their CHD4 data a lot with previous FRET data on CHD1, therefore it might be good to summarize those already in the introduction and by that raise the question to be answered in this study on CHD4.
- The gel based data is beautiful. All gel based experiments were performed at 37°C, while the smFRET assays were performed at room temperature. I assume that the time scale of remodeling changes strongly with these 15K difference. Could the authors perform one set of gel based assays e.g. Figure 1D at room temperature? These would be important for the later on interpretation of smFRET kinetics.
- From their pull down assays the authors find in Figure 2 affinities of 40-80nM (N.B. one closing bracket is missing at the first statement). In their discussion the authors claim affinities of 30-40nM. Could the authors please explain this difference? Would it make sense to extract and plot lane intensities from the EMSA to quantify the affinity?
- The design and presentation of their exit side smFRET experiments is very elegant. The placement of fluorophores apparently did not hinder CHD4 dynamics. The authors find that CHD4 activity releases DNA in 6 and 4 bp steps (page 6, last paragraph). Did the authors analyze (and in how many molecules) that it was always first a 6bp step followed by a 4 bp step? Can the authors reason why it is this way and not the other way? I would suggest to build a histogram of step sizes (first step and second step). This can yield important information of the molecular mechanism of how torsion propagates around the histone-octamer.
- I am more puzzled with the entry site experiments. The authors show that the protein in absence of ATP can already remodel the DNA at the entry side. Is the FRET change of 0.2 to 0.6 reflecting only a 1 bp step? if not, where does the huge amount of energy come from for turning and buckling a 6 bp dsDNA loop? I assume in absence of ATP it is a 1-3 bp shift only, simply due to energetic reasons.
- It remains very unclear how the actual smFRET experiment was performed. The authors describe that in their experiments they used a microfluidic system for CHD4 injection. I then assume that the time given in the FRET efficiency traces is a true time. While in the exit experiments the authors see within 100s (1mM ATP) clear remodeling activity, in the entry experiments with CHD4 on its own the increase in FRET for maybe a 1-3 bp shift takes 50 s or longer and is often reversed. How do the authors combine both experiments?
- I think it would be important to determine here the reaction dependence on the CHD4 concentration. I am aware that the actual movement is independent of CHD4 concentration (single CHD4 activity), but from injection to start of the reaction there should be a lag time depending on the protein concentration. This rate of initiation could be determined from entry or exit flow experiments. This together with the exit kinetics could help to model the entry kinetics further.
- according to the authors explanation the entry experiments might be strongly affected by the fluorophore labeling. Wouldn't it be possible to label the DNA at position +4 and -4 where the DNA should move relativ to each other in CHD4 action and use this read out as a further evidence that

the DNA is moved around the octane in these experiments in a kind of reptation mode (referring to SSB movement found by Ha and colleagues).

- coming back to the temperature effects. Did the authors incubate CHD4 in the smFRET experiment at the entry side experiments for 10 or 20 min without illumination and then probe the FRET efficiency afterwards? Also, it might be possible to obtain kinetic information of this slow process by snap illumination every min for a few frames and by that track the nucleosome to still perform a true single molecule experiment

- I hardly understand the color code presented in Figure 7. What do the red strands indicate, and in Figure 7F if the stress is released for a forward movement, why is there again a lot of DNA at the backwards end present?

- CHD4 is huge! nearly 2000aa. The Figure 7 suggests otherwise a tiny little protein occupying only 3-4 bp of dsDNA. This is particularly irritating when the authors argument that their fluorophore at the entry side was hindering entry, although this seems to be tens of bp away. Maybe a three dimensional depiction based on Figure 1A would be more helpful?

Minor:

- Figure 3F: it would be great to give the slope a unit of -0.051 deltaE/bp ?

Reviewer #2 (Remarks to the Author):

In this manuscript, Zhong & Paudel et al. combined bulk and single-molecule in vitro biochemical assays to investigate the mechanism of human CHD4 chromatin remodeler, one of the subunits of NuRD remodeling/deacetylase complex which is critical for regulation of gene expression. This study continues a string of reports published by the Mackay lab on the CHD4 remodelling mechanism. The authors find that CHD4 remodels mono-nucleosomes by de-coupled ATP-dependent translocation of nucleosomal DNA at the entry and exit sides, with gradual DNA translocation at entry side and abrupt stepwise translocation at the exit side. Overall, the manuscript is well-written, and the experiments are designed properly in most of the cases. I recommend publishing this study in Nature Communications, after addressing some minor points that I will discuss below.

• Page 3 (lines 14 – 17) & Fig. 1C

The authors claim that remodeling activity of isolated CHD4 is identical with corresponding activity of full NuRD complex. Regarding the amounts of proteins (CHD4 vs. NuRD) compared, the authors mentioned in the Methods section that they estimated the concentrations based on densitometry of Sypro Ruby stained SDS-PA gel, but it is not clear how they correlated image density to protein amount. Did they use calibration curve based on purified reference protein, like BSA?

The authors claim that NuRD complex and CHD4 have identical reaction yield or extend, but band intensities are not really the same. Can the authors please clarify this?

The authors compared NuRD and CHD4 remodelling activities using less preferred nucleosome substrate (asymmetric). Since CHD4 prefers symmetric nucleosome substrate, then why did the authors not use it to compare CHD4 and NuRD remodelling activities? Does NuRD complex possess the same substrate preference?

• Page 3 (lines 31 – 33) & Fig. 1E

Based on results shown in Fig. 1-E, the authors claim that CHD4 can discriminate between substrates with different DNA sequences. It is hard for me derive this conclusion from the gel image provided. Also, I think it will be more convincing to test nucleosomes assembled with different other positioning DNA sequences.

• Page 3 (lines 37 – 38) & Fig. 1F

The authors claim that CHD4 can remodel nucleosomes with a 30-bp extension based on data shown in Fig. 1-F of the 0w30 and 30w0 substrates, however I cannot see any detectable products at position higher than the position of initial reactant substrate. Please clarify.

- Page 5 (line 1) & Fig. 2

The authors main conclusion out of all data shown in Fig's 2A and 2B is that affinity of CHD4 for nucleosomes is independent of length of flanking DNA. What I can see in Fig. 2A that there is more binding with 0w60 than with 0w0. Also, there is more binding with 60w60 than with 30w30. In addition, I see in Fig. 2B that more nucleosomes bound to CHD4 in cases of 0w30 and 0w60 compared to 0w0. Besides, more nucleosomes bound to CHD4 in cases of 0w60 compared to 0w30. Overall, I see from the data that there is dependence on length of flanking DNA.

- Fig. 3 B and C

I could not find anywhere in the manuscript or the SI files any representative single-molecule traces from distal-only fluorescent nucleosomes or from doubly labelled fluorescent nucleosomes with two steps of photobleaching from two acceptor dyes on one nucleosome. I would suggest including those traces at least in the SI file.

- Fig. 3F

I think the x-axis should be labelled as 'length of linker DNA', not nucleosomes.

- Fig. 4B (lowermost)

No pause can be observed in the 20 mM ATP trace (lowermost) that the authors chose to show in this figure. Although at this high ATP concentration pauses are expected to be the shortest, but I recommend the authors to make it clearer in the figure.

- Fig. 4D (lower)

In the caption, it is written as 'N = 1 - 3', but I think it should be 'N = 1 - 5'.

Referee 1

The introduction of this manuscript appears cluttered at the moment. After reading it several times, I am still not sure if I can follow it. In the discussion the authors compared their CHD4 data a lot with previous FRET data on CHD1, therefore it might be good to summarize those already in the introduction and by that raise the question to be answered in this study on CHD4.

- We have significantly altered the text of the Introduction to address these issues. We have included a summary of key published smFRET data on chromatin remodelling enzymes, including work done on CHD1. This new text is highlighted in yellow.

The gel based data is beautiful. All gel based experiments were performed at 37°C, while the smFRET assays were performed at room temperature. I assume that the time scale of remodeling changes strongly with these 15K difference. Could the authors perform one set of gel based assays e.g. Figure 1D at room temperature? These would be important for the later on interpretation of smFRET kinetics.

- We have carried out a gel-based assay at room temperature. The remodelling behaviour, in terms of the bands that are observed and their relative intensities, is essentially unchanged from the 37 °C data shown in the original manuscript. This new gel is shown as **Supplementary Figure 1A**.

From their pull down assays the authors find in Figure 2 affinities of 40-80nM (N.B. one closing bracket is missing at the first statement). In their discussion the authors claim affinities of 30-40nM. Could the authors please explain this difference? Would it make sense to extract and plot lane intensities from the EMSA to quantify the affinity?

- This is a typo – it should be 40-80 in both cases. We have corrected this.

The design and presentation of their exit side smFRET experiments is very elegant. The placement of fluorophores apparently did not hinder CHD4 dynamics. The authors find that CHD4 activity releases DNA in 6 and 4 bp steps (page 6, last paragraph). Did the authors analyze (and in how many molecules) that it was always first a 6bp step followed by a 4 bp step? Can the authors reason why it is this way and not the other way? I would suggest to build a histogram of step sizes (first step and second step). This can yield important information of the molecular mechanism of how torsion propagates around the histone-octamer.

- We have constructed histograms for the 1st and 2nd steps at various ATP concentrations. These are shown in **Figure 3F** and **Supplementary Figure 1G**, and the peaks of their distribution are consistently being 6 and 4 bp, respectively. The distribution of the 2nd drop was wider than that of the first and thus indicated a greater variation. At this stage, we do not have a mechanistic explanation for the observed difference in step size. It is possible that the overall state of the nucleosomal DNA is no longer the same as the initial state after one round of remodelling, and might therefore be more prone to the stress created by the continuous twist of DNA at the entry site.

I am more puzzled with the entry site experiments. The authors show that the protein in absence of ATP can already remodel the DNA at the entry side. Is the FRET change of 0.2 to 0.6 reflecting only

a 1 bp step? if not, where does the huge amount of energy come from for turning and buckling a 6 bp dsDNA loop? I assume in absence of ATP it is a 1-3 bp shift only, simply due to energetic reasons.

- The referee has interpreted the FRET change in the entry-side experiment (for distally labelled molecules) by using the calibration curve that we measured for exit-side (proximally labelled) molecules. However, this is not valid because the chemical environments of these two setups and the method of labelling are different. Our view is in line with that of the referee, in that we propose that the fluorophore is likely to shift by ~1–3 base pairs. Note that it is also possible that changes in orientation of the fluorophore can affect the observed FRET, further complicating a quantitative analysis.

It remains very unclear how the actual smFRET experiment was performed. The authors describe that in their experiments they used a microfluidic system for CHD4 injection. I then assume that the time given in the FRET efficiency traces is a true time. While in the exit experiments the authors see within 100s (1mM ATP) clear remodeling activity, in the entry experiments with CHD4 on its own the increase in FRET for maybe a 1-3 bp shift takes 50 s or longer and is often reversed. How do the authors combine both experiments?

- The time given in the FRET traces is indeed a true time.
- The entry- and exit-side experiments cannot be directly combined because each particle behaves independently. Overall, the changes are happening on similar timescales, consistent with our interpretation. It is also not possible to quantitatively interpret the initial time before the first change in FRET because of the nonlinear nature of flow in the microfluidic flow cells. It is for this reason that our analysis has been restricted to the time between FRET changes.

I think it would be important to determine here the reaction dependence on the CHD4 concentration. I am aware that the actual movement is independent of CHD4 concentration (single CHD4 activity), but from injection to start of the reaction there should be a lag time depending on the protein concentration. This rate of initiation could be determined from entry or exit flow experiments. This together with the exit kinetics could help to model the entry kinetics further.

- First of all, the starting times are measured from the time the CHD4 is injected. This is followed by a significant delay time (probably the majority of the delay time we see) before it physically enters the flow cell. Because of fluid dynamics (as noted above), the CHD4 will not arrive in a coherent manner at the slide surface, but rather will show up as a gradually increasing concentration, making it impossible to measure rate constants in an unambiguous manner. Furthermore, the microfluidic device varies from day to day in ways that affect flow rate. Therefore, we believe that it is not practical to measure meaningful initial binding kinetics.

According to the authors explanation the entry experiments might be strongly affected by the fluorophore labeling. Wouldn't it be possible to label the DNA at position +4 and -4 where the DNA should move relative to each other in CHD4 action and use this read out as a further evidence that the DNA is moved around the octane in these experiments in a kind of reptation mode (referring to SSB movement found by Ha and colleagues).

- We would be very keen to obtain more information on the remodelling mechanism by introducing fluorophores at other positions on the DNA. However, our data (both FRET and gel-based remodelling) indicate that the presence of covalently attached fluorophores severely

inhibits remodelling, and so we predict that labelling in the manner suggested by the referee would unfortunately perturb or abrogate remodelling.

Coming back to the temperature effects. Did the authors incubate CHD4 in the smFRET experiment at the entry side experiments for 10 or 20 min without illumination and then probe the FRET efficiency afterwards? Also, it might be possible to obtain kinetic information of this slow process by snap illumination every min for a few frames and by that track the nucleosome to still perform a true single molecule experiment

- In our experiments, we track the reaction for 3-5 mins; this time scale is sufficient for remodelling.

I hardly understand the color code presented in Figure 7. What do the red strands indicate, and in Figure 7F if the stress is released for a forward movement, why is there again a lot of DNA at the backwards end present? CHD4 is huge! nearly 2000aa. The Figure 7 suggests otherwise a tiny little protein occupying only 3-4 bp of dsDNA. This is particularly irritating when the authors argument that their fluorophore at the entry side was hindering entry, although this seems to be tens of bp away. Maybe a three dimensional depiction based on Figure 1A would be more helpful?

- It is true that CHD4 is very large – larger than depicted in this diagram. However, the diagram only depicts the ATPase domain of CHD4, which is approximately a third of the size of the full protein. We have also added a supplementary video to depict the mechanism in a three-dimensional way.

Minor:

Figure 3F: it would be great to give the slope a unit of -0.051 deltaE/bp ?

- We have fixed this.

Referee 2

Overall, the manuscript is well-written, and the experiments are designed properly in most of the cases. I recommend publishing this study in Nature Communications, after addressing some minor points that I will discuss below.

Page 3 (lines 14 – 17) & Fig. 1C. The authors claim that remodeling activity of isolated CHD4 is identical with corresponding activity of full NuRD complex. Regarding the amounts of proteins (CHD4 vs. NuRD) compared, the authors mentioned in the Methods section that they estimated the concentrations based on densitometry of Sypro Ruby stained SDS-PA gel, but it is not clear how they correlated image density to protein amount. Did they use calibration curve based on purified reference protein, like BSA?

- Yes. CHD4 and NuRD were loaded onto a gel along with a known amount of BSA and stained with SyproRuby; concentrations were then estimated using Image J. We have clarified in the Methods section.

The authors claim that NuRD complex and CHD4 have identical reaction yield or extend, but band intensities are not really the same. Can the authors please clarify this?

- We believe that these band intensities **are** essentially the same, within the limits of reproducibility of these gel-based remodelling assays.

The authors compared NuRD and CHD4 remodelling activities using less preferred nucleosome substrate (asymmetric). Since CHD4 prefers symmetric nucleosome substrate, then why did the authors not use it to compare CHD4 and NuRD remodelling activities? Does NuRD complex possess the same substrate preference?

- The focus of the manuscript is on the behaviour of CHD4, rather than the full NuRD complex. The experiment suggested by the referee is an important one for understanding NuRD remodelling activity, and this will be the focus of an upcoming manuscript. **Figure 1C** demonstrates that there are no dramatic differences between CHD4 and NuRD in their activity on the nucleosome substrate that is most commonly used in all studies of nucleosome sliding activity.

Page 3 (lines 31 – 33) & Fig. 1E. Based on results shown in Fig. 1-E, the authors claim that CHD4 can discriminate between substrates with different DNA sequences. It is hard for me derive this conclusion from the gel image provided. Also, I think it will be more convincing to test nucleosomes assembled with different other positioning DNA sequences.

- To address this concern, we have carried out gel-based remodelling experiments with the MMTV nucA positioning sequence¹. Since it is a native sequence, it does not form a uniquely positioned nucleosome after reconstitution, but rather a dominant band with fainter bands representing nucleosome positioned at other sites. After remodelling, the relative intensities of the bands change significantly (**Supplementary Figure 1B**). This effect was more obvious with 60w0 MMTV nucleosomes, and is consistent with the published observation of MMTV nucA nucleosome remodelling².

Page 3 (lines 37 – 38) & Fig. 1F. The authors claim that CHD4 can remodel nucleosomes with a 30-bp extension based on data shown in Fig. 1-F of the 0w30 and 30w0 substrates, however I cannot see any detectable products at position higher than the position of initial reactant substrate. Please clarify.

- We do see a band. It's clear in the 30w0 case though not so clear in the 0w30 case.

Page 5 (line 1) & Fig. 2. The authors main conclusion out of all data shown in Fig's 2A and 2B is that affinity of CHD4 for nucleosomes is independent of length of flanking DNA. What I can see in Fig. 2A that there is more binding with 0w60 than with 0w0. Also, there is more binding with 60w60 than with 30w30. In addition, I see in Fig. 2B that more nucleosomes bound to CHD4 in cases of 0w30 and 0w60 compared to 0w0. Besides, more nucleosomes bound to CHD4 in cases of 0w60 compared to 0w30. Overall, I see from the data that there is dependence on length of flanking DNA.

- We agree that the intensities of the histone bands are not identical in the different lanes of **Figure 2A**. However, we did not design this experiment to be quantitative – rather, it is simply included to demonstrate that CHD4 **can** bind to all of these nucleosomal substrates.

- It is also true that the band intensities are not identical in **Figure 2B** across the different substrates. However, the differences in intensity are consistent with differences in dissociation constant with a factor of ~ 2 , and we would not want to infer dissociation constants from gel shift data with a greater precision than that because of the other factors that can influence band intensity. For example, the appearance of bands in a gel shift is dependent both on the affinity of the complex and the kinetics of dissociation of the complex, complicating affinity comparisons. It is for this reason that we state that the KD values are within a factor of ~ 2 of each other and avoid overinterpreting the data.

Fig. 3 B and C. I could not find anywhere in the manuscript or the SI files any representative single-molecule traces from distal-only fluorescent nucleosomes or from doubly labelled fluorescent nucleosomes with two steps of photobleaching from two acceptor dyes on one nucleosome. I would suggest including those traces at least in the SI file.

- We have included two traces showing two doubly labelled nucleosomes undergoing two steps of photobleaching as **Supplementary Figure 1C**. One trace had the proximal fluorophore photobleached followed by the distal label, whereas the other had the distal label bleached first. The text has also been amended to reflect this change (Page 6, line 33).

Fig. 3F. I think the x-axis should be labelled as 'length of linker DNA', not nucleosomes.

- We have corrected the legend.

Fig. 4B (lowermost). No pause can be observed in the 20 mM ATP trace (lowermost) that the authors chose to show in this figure. Although at this high ATP concentration pauses are expected to be the shortest, but I recommend the authors to make it clearer in the figure.

- We have amended **Figure 4B** to address the reviewer's comment, showing the very short pause that is observed at this ATP concentration.

Fig. 4D (lower). In the caption, it is written as ' $N = 1 - 3$ ', but I think it should be ' $N = 1 - 5$ '.

- We have corrected this legend.

Reference

- 1 Richardfoy, H. & Hager, G. L. Sequence-Specific Positioning of Nucleosomes over the Steroid-Inducible Mmtv Promoter. *Embo J* **6**, 2321-2328, doi:DOI 10.1002/j.1460-2075.1987.tb02507.x (1987).
- 2 Stockdale, C., Flaus, A., Ferreira, H. & Owen-Hughes, T. Analysis of nucleosome repositioning by yeast ISWI and Chd1 chromatin remodeling complexes. *J Biol Chem* **281**, 16279-16288, doi:10.1074/jbc.M600682200 (2006).

Reviewers' comments:

Reviewer #1 (Remarks to the Author):

Zhong, Paddle and colleagues present here a revised manuscript of their study of the nucleosome remodeller CHD4. Overall, I am mostly satisfied with their revision. The manuscript reads now much more easy and the conclusions appear rather justified - even though I would have liked to see more of the additional experiments included to learn more about the entry site dynamics and the correlation between both. There are a few minor points I would like to see addressed, mainly from a technical side:

- In the introduction, the authors introduced a new, well written paragraph on page two. Here, there is a small typo with ISW/SNF instead of SWI/SNF. Please change.

- I am not entirely sure how the histograms wer generated and plotted in Figure 4D. Typically software puts the sum to the top left corner of a histogram bar, however, the better representation would be a bin-centred representation. Please check if this was used in Figure 4, then the gamma distribution fits also would be a much better fit to the data.

- I am unsure about Figure 4C vs Figure 4D. To my understanding the top histograms are the same data, however, the histogram in 4D has a much longer tail. Please clarify this.

- Regarding the pauses the authors use t_{pause} and τ_{pause} . Please harmonise in figures and legends and text.

- I am aware that it is much easier to talk about FRET. However, what is measured is a FRET efficiency and a change in FRET efficiency. Could the authors please adjust this small but technical important difference in their figures and text. E.g. E_{FRET} as a symbol and thus E_{FRET} change instead of FRET and FRET change.

- I feel that I saw the E_{FRET} calibration in this manuscript already before in a manuscript by the Ha lab. It might be worth crediting this?

One last remark - maybe semantic - but important for science IMHO:

- In my previous report I noted that the authors reported different affinities of 40-80 nM and 30-40 nM. The authors claimed this is a "typo". I just want to clarify that this is not what I would call a "typo" (see first point above, this is a typo), but it was just wrong in one place, and it is fine to admit a mistake and correct it, but putting this down as a typo is giving a wrong impression about solidity of research.

Reviewers' comments:

Reviewer #1 (Remarks to the Author):

- In the introduction, the authors introduced a new, well written paragraph on page two. Here, there is a small typo with ISW/SNF instead of SWI/SNF. Please change.

We have changed this.

- I am not entirely sure how the histograms were generated and plotted in Figure 4D. Typically software puts the sum to the top left corner of a histogram bar, however, the better representation would be a bin-centred representation. Please check if this was used in Figure 4, then the gamma distribution fits also would be a much better fit to the data.

The fit was originally to the top left corner of the histogram bars. We have now changed figure 4C and 4D to be a bin-centred representation, as suggested by the reviewer.

- I am unsure about Figure 4C vs Figure 4D. To my understanding the top histograms are the same data, however, the histogram in 4D has a much longer tail. Please clarify this.

Figure 4C and 4D are generated using same data and a careful inspection will show that they are same histograms. The histogram in Figure 4D therefore does not (cannot) have longer tail. Perhaps it appears this way to the referee because of the different aspect ratios used in the two plots?

It is also possible that the referee perceives the two histograms to be different because the N=1 fitting curve accidentally connects the two bars flanking the gap at ~120 s and makes it look like an extra bar (the figure below shows the N=1 curve with two different transparency and thus the gap between the two bars, indicated by the yellow circle, can be seen). We have made the strokes thinner to address this ambiguity in the revised manuscript.

- Regarding the pauses the authors use t_{pause} and τ_{pause} . Please harmonise in figures and legends and text.

We have changed all τ_{pause} to t_{pause} .

- I am aware that it is much easier to talk about FRET. However, what is measured is a FRET efficiency and a

change in FRET efficiency. Could the authors please adjust this small but technical important difference in their figures and text. E.g. E_FRET as a symbol and thus E_FRET change instead of FRET and FRET change.

We agree with the reviewer that it is FRET efficiency that is being measured. We have therefore added a sentence in cyan the Results section (second paragraph in the subsection “CHD4 ejects DNA from the nucleosome in multi-base-pair steps”) to clarify that when we write FRET, we are referring to FRET efficiency.

- I feel that I saw the E_FRET calibration in this manuscript already before in a manuscript by the Ha lab. It might be worth crediting this?

Yes. We have added the reference [1], in which they constructed a similar calibration curve.

Reference:

1. Blosser, T.R., et al., *Dynamics of nucleosome remodelling by individual ACF complexes*. Nature, 2009. **462**(7276): p. 1022-U79.

REVIEWERS' COMMENTS:

Reviewer #1 (Remarks to the Author):

The authors have addressed all my remaining questions.